# Computational Procedure for Analysis of Crystallites in Polycrystalline Solids of Quasilinear Molecules

**DOI:** 10.3390/molecules28052327

**Published:** 2023-03-02

**Authors:** Stoyan Iliev, Sonya Tsibranska, Ilia Kichev, Slavka Tcholakova, Nikolai Denkov, Anela Ivanova

**Affiliations:** 1Department of Physical Chemistry, Faculty of Chemistry and Pharmacy, University of Sofia, 1 James Bourchier Blvd., 1164 Sofia, Bulgaria; 2Department of Chemical and Pharmaceutical Engineering, Faculty of Chemistry and Pharmacy, University of Sofia, 1 James Bourchier Blvd., 1164 Sofia, Bulgaria

**Keywords:** structural analysis, separation of crystallites, crystallite eigenplane, molecular tilt, hexadecane-surfactant interface, molecular dynamics

## Abstract

In the current work, a comprehensive procedure for structural analysis of quasilinear organic molecules arranged in a polycrystalline sample generated by molecular dynamics is developed. A linear alkane, hexadecane, is used as a test case because of its interesting behavior upon cooling. Instead of a direct transition from isotropic liquid to the solid crystalline phase, this compound forms first a short-lived intermediate state known as a “rotator phase”. The rotator phase and the crystalline one are distinguished by a set of structural parameters. We propose a robust methodology to evaluate the type of ordered phase obtained after a liquid-to-solid phase transition in a polycrystalline assembly. The analysis starts with the identification and separation of the individual crystallites. Then, the eigenplane of each of them is fit and the tilt angle of the molecules relative to it is computed. The average area per molecule and the distance to the nearest neighbors are estimated by a 2D Voronoi tessellation. The orientation of the molecules with respect to each other is quantified by visualization of the second molecular principal axis. The suggested procedure may be applied to different quasilinear organic compounds in the solid state and to various data compiled in a trajectory.

## 1. Introduction

Upon cooling, alkanes and their mixtures can form intermediate structural phases between the isotropic liquid and the regularly ordered solid crystalline phase [1,2]. These transient phases are known in the literature as “rotator phases” or “plastic phases”. The names stem from the relative freedom of the molecules to rotate about their long axis. At the same time, these structures possess long-range positional order in three-dimensional space, similar to that in a real crystal lattice. Rotator phases are typical for middle-length (above 14 carbon atoms) and longer alkyl chains [3]. Experiments reveal five possible rotator phases (denoted R_I_ to R_V_ in the literature) [2]. They are discerned by a number of order parameters. Some of them define the shape of the unit cell—a deformed hexagon, the tilt angle of the molecules with respect to the crystal plane (eigenplane), and the orientation of the molecules with respect to each other [2]. When in the rotator phase, the materials exhibit complex visco-plastic rheological behavior [4], which may be employed to control the shape and size of alkane-based nanoassemblies.

While the rotator phases are studied well experimentally [3], there are still many unanswered questions at the molecular level, related to the mechanism of their formation and to the intermolecular orientation therein, due to insufficient resolution of the experimental techniques. Modeling of the freezing or melting of alkane-based systems with molecular dynamics (MD) can provide insight into the nature of the formation of the intermediate phases. Such simulations may also describe transformations between transient phases, which are too short-lived to be observed experimentally.

Some molecular simulation studies focused on rotator phases investigate the thermodynamic characteristics of the systems. Rao et al. [5,6] calculated the diffusion coefficient and the heat capacity (*C_p_*) of single-component alkane systems and binary mixtures of alkanes. Tsuchiya et al. [7,8] developed a procedure to predict the melting point of n-alkanes utilizing radial distribution functions (RDF) and to calculate the latent heat of the phase transition.

Other theoretical works are dedicated to determining the structural characteristics of rotator phases and of the transitions between these phases. Visual observations and order parameters dependent on the *gauche*/*trans* ratio of dihedrals along the alkyl chains are often employed [9,10,11,12,13,14]. Cao et al. [15] developed a parameter to monitor the orientational order upon rotation about the chain axes in a crystal or in a rotator phase. However, it depends on the initial structure of the system. Milner et al. [16] studied the molecular conformation, mobility, formation energy, and fluctuation-free energy of twist solitons. To track the orientation of twists in the soliton, they suggested an order parameter indicating the orientation of the atoms in the molecule. However, this parameter is local in nature.

In addition to quantifying the *gauche*/*trans* ratio, Ryckaert et al. constructed a matrix from the coordinates of vectors representing the molecules in the crystal. The eigenvector corresponding to the highest eigenvalue of this matrix was employed as a measure of the average tilting of the molecules in the crystal with respect to the *z* axis of the coordinate system. Furthermore, they calculated an additional parameter that quantified the orientation of the molecules with respect to the *x* axis of the coordinate system. Both analyses rely on the crystal plane being parallel to the *xy* plane of the coordinate system. Thus, in the case of a crystal randomly oriented in space, the applicability of these analyses would be limited.

Wentzel et al. [17] studied the transitions of alkanes from R_I_ to R_II_ and to crystal phases. To distinguish between the two phases, they developed Ising- and Potts-like order parameters. These parameters describe well the local ordering in the crystal. However, if there are disordered parts in the crystal, an additional correction needs to be made. This post-processing complicates the analysis of systems with multiple ordered domains. Moreover, the calculation of the two order parameters relies on the molecules not diffusing. Burrows et al. [18] calculated the second principal axis for each molecule in an alkane crystal to determine the orientation of molecules with respect to each other. This approach allows easy and fast evaluation of the intermolecular orientation in the system. However, the long molecular axes of the molecules are assumed to be parallel to one of the axes of the coordinate system, which might not be the actual case.

Most of the works published so far use a single structural parameter to identify the type of solid phase formed. However, determining the phase state of an alkane crystal requires the evaluation of multiple structural properties that vary between the different rotator phases. Moreover, most alkane solids have a polycrystalline structure with crystallites oriented in different directions in space. Therefore, a more comprehensive procedure for the analysis of alkanes (or other quasilinear compounds) in the solid state is needed. It should allow identification and separation of all crystallites and unequivocal assignment of the type of solid-state packing therein.

In the current study, we devise a trajectory-based systematic analysis protocol for alkane-containing assemblies in the solid state, which fulfills these objectives. Crystallites with different degrees of ordering may be separated efficiently and analyzed individually. The assignment of the phase state is done on the basis of a consistent set of analyses that include fitting of the eigenplane of each crystallite, calculation of the evolution of the tilt angle of the molecules relative to it, as well as the time-averaged tilt angle for each molecule, radial distribution functions, and 2D Voronoi analysis. We extend also the second principal axis analysis to be applicable to molecules not parallel to one of the directions of the coordinate system. The procedure is generally pertinent to various solid-state structures (ranging from liquids through single crystals to polycrystalline assemblies) of quasilinear molecules with representative configurations compiled in a simulation trajectory file.

The MD data analyses shown in this work are aimed at illustrating the developed methodology. More details about the construction of the models and the MD simulations may be found in our recent publication [19]. Therein, in-depth structural analyses of the characteristics of hexadecane-containing systems are made by using the procedures developed in the current study.

## 2. Results and Discussion

### 2.1. Identification and Separation of the Different Crystallites in the System

The crystallites obtained in the systems bulk HEX and HEX/Surf/water (see Section 3 for description of the models) after their freezing are illustrated in Figure 1. It is evident that each crystallite at 278 K points in a different direction in space. By contrast, the HEX/Surf/water system at 289 K forms a single crystal with all ordered molecules aligned in one direction. The reason, however, is not the different final temperature but the fact that a nucleation seed was present in the initial configuration of the simulation at 289 K. This result demonstrates that the freezing point obtained in the simulations is different from the experimental value due to the supercooling effect stemming from the nanodimensions of the model systems. Concurrently, the single crystal formed at 289 K offers a structural pattern in good contrast with the polycrystalline samples, which may be used to test the broader applicability of the analysis procedure.

Because of the nature of the obtained crystallites (Figure 1), one can use it as a criterion for isolating a crystallite and its orientation in space. To characterize this orientation, we construct a vector for each HEX molecule and/or surfactant alkyl tail spanning the third (*C_3_*) and fourteenth (*C_14_*) carbon atoms (Figure 2A).

These atoms are chosen as they are relatively close to the termini of the molecule, while being more rigid compared to the terminal methyl groups. Then, we calculate the slopes *B_ij_* of the projections of the $\vec{C_{3}C_{14}}$ vector of each molecule in the *xy*, *yz,* and *zx* planes of the coordinate system:$$B_{xy}^{k}=\arctan\frac{\Delta y^{k}}{\Delta x^{k}}=\arctan\frac{y_{C_{14}}^{k}-y_{C_{3}}^{k}}{x_{C_{14}}^{k}-x_{C_{3}}^{k}}$$
(1)$$B_{yz}^{k}=\arctan\frac{\Delta z^{k}}{\Delta y^{k}}=\arctan\frac{z_{C_{14}}^{k}-z_{C_{3}}^{k}}{y_{C_{14}}^{k}-y_{C_{3}}^{k}}$$
$$B_{zx}^{k}=\arctan\frac{\Delta x^{k}}{\Delta z^{k}}=\arctan\frac{x_{C_{14}}^{k}-x_{C_{3}}^{k}}{z_{C_{14}}^{k}-z_{C_{3}}^{k}}$$

The index *k* denotes the *C_3_* and *C_14_* positions of the *k^th^* molecule.

The slopes are averaged over a certain time period. The latter is selected to be at the end of the trajectory or to correspond to an interval in which the system is in a stationary state. In our case, the last 50 ns of each trajectory were chosen.

Plotting the average slopes on a 3D graph (Figure 2B–D) results in well-separated clusters of points, each corresponding to a single crystallite.

Each cluster is identified numerically by the ranges of the slopes it spans along the three axes. The points that are scattered in between the clusters correspond to molecules that are still disordered, i.e., they do not belong to an ordered domain. It should be noted that in the limiting case of a very well-ordered single crystal, its slope will also appear on the graph as a small cluster. Then, standard deviations could be used to discriminate unequivocally between the disordered molecules and the single crystal (Figure 2D).

It should be mentioned that in the limiting case where the vector $\vec{C_{3}C_{14}}$ lies in one of the planes of the coordinate system, this procedure could become numerically unstable. This is due to the fact that one of the slopes is close to zero which results in a tangent of ±∞. This obstacle can be overcome by calculating the angles of $\vec{C_{3}C_{14}}$ with the three axes of the coordinate system (Equation (2)).
$$\alpha_{x}^{k}=\arccos\frac{x_{C_{14}}^{k}-x_{C_{3}}^{k}}{\vec{|C_{3}C_{14}}|}$$
(2)$$\alpha_{y}^{k}=\mathrm{arcos}\frac{y_{C_{14}}^{k}-y_{C_{3}}^{k}}{\vec{|C_{3}C_{14}}|}$$
$$\alpha_{z}^{k}=\arccos\frac{z_{C_{14}}^{k}-z_{C_{3}}^{k}}{\vec{|C_{3}C_{14}}|}$$

After the separate crystallites in the trajectory are isolated, they often appear split or look as if they contain vacancies when presented in the simulation box (Figure 3A,B).

For some of the analyses described later in the paper, for example in the procedure aiming to determine the eigenplane of the crystallite, this apparent crystallite fragmentation is detrimental. Hence, a procedure to reconstruct the real-space coordinates of the whole crystallite from the periodic images was developed. It is applied separately on each crystallite.

### 2.2. Reconstructing the Real-Space Coordinates of the Crystallite

Crystallites, which appear split and need to be imaged across the periodic box to reconstruct their actual coordinates, always have a void between the centers of mass (COM) of the molecules located in different regions of the periodic box (Figure 3A). In our approach, we detect these voids and measure the distance between each pair of slabs to identify the molecules, which need to be moved. After that, we image these groups of molecules along one of the axes of the periodic box to obtain the real-space coordinates. A graphical illustration of this procedure is presented in Figure 3A.

To evaluate the voids between the different slabs of molecules, we rotate the crystallite such that the long axes of the molecules therein are aligned with the *z* axis of the coordinate system. In this way, the *z* coordinate of the COM of the molecule determines the slab to which a molecule belongs because maximum intermolecular separation along *z* is achieved.

First, the *z* coordinates of the COM of all molecules in the crystallite are sorted in ascending order. Then, a dictionary is created in the format: key, list of values. The COM *z*-position of the first molecule (with the lowest *z* coordinate) in each slab is used as a key. The list associated with it contains the molecule IDs (resids) of all molecules belonging to this slab. The end of a given slab is determined by a comparison of the difference in the COM *z* coordinates of adjacent pairs in the sorted array. Whenever this difference becomes larger than a predefined criterion for a void (specific for each system), the slab terminates. Then, a new key is created for the next slab and the next list is started.

When all molecules are assigned to slabs, the keys of the separate slabs are indexed from 0 to N-1, e.g., the four slabs illustrated in Figure 3A will have indices 0, 1, 2, and 3. The completed dictionary is then used to evaluate how to image the molecules in each slab so that their real-space coordinates are reconstructed. The direction of imaging, i.e., *x*, *y*, or *z*, is determined visually by the user because it depends on the crystallite orientation in the box. The length of the translation vector along this direction is always a multiple (*kL_x/y/z_*) of the periodic box size in the same direction (*L_x/y/z_*). For the example shown in Figure 3B the imaging vector magnitude is *−1L_y_*. The magnitude of *k* is calculated from the difference of the indices of the keys. One of the slabs, e.g., the one containing the largest number of molecules (colored in green in Figure 3A), is selected as a reference. Then, the differences between the index of its key and those of the other slabs are calculated. The obtained numbers are the values of the parameter *k* for the respective slab. For example, the blue slab in Figure 3A has to be imaged by 2*L_x_* in the *+x* direction, whereas the cyan cluster is imaged once along *–x*, i.e., *k* = −1.

The imaging procedure is repeated for each frame in the trajectory file. It should be noted that the imaging is applied on the non-rotated original trajectory to preserve the periodicity of the crystallite. After the real-space crystallite is properly reconstructed, its eigenplane is calculated as the next step of the analysis.

The procedure described in this section is applied only when the crystallite is split in order to fit in the periodic box. This is determined from the final location of the imaged molecules—they are imaged only if filling vacancies in the image layer (Figure 3A). If the crystallite has an actual multilayer structure, it is preserved.

### 2.3. Fitting of the Eigenplane of a Crystallite and Calculating the Tilt Angle

To calculate the crystallite eigenplane, we compute the covariance matrix of the *x*, *y*, and *z* coordinates of an atom in the middle of the chain for each molecule in the crystallite—namely atom *C_8_* (Figure 2A) in the case of hexadecane or the surfactant tail. The coordinate matrix **M** has the form:(3)$$M=\left(\begin{array}{cccc} x_{1} & x_{2} & \ldots & x_{n}\\ y_{1} & y_{2} & \ldots & y_{n}\\ z_{1} & z_{2} & \ldots & z_{n} \end{array}\right),$$
where each column contains the three coordinates of the atom from each molecule and *n* is the total number of molecules in the crystallite. The covariance matrix is computed as:(4)$$MM^{T}$$

The matrix in Equation (4) is similar to the one used by Ryckaert et al. (denoted as Q in the publication) [11]. There, the eigenvector with the largest eigenvalue is employed to calculate the average molecular tilt with respect to the *z* axis of the Cartesian coordinate system. In our case, however, we determine the eigenplane of the crystallite as described below. In a subsequent step of our analysis, we estimate the tilt angle of the molecules with respect to this eigenplane.

Calculating the eigenvectors of the covariance matrix [20] gives an orthogonal basis, where the vector corresponding to the lowest eigenvalue is the normal to the plane of the crystallite. This vector ($\vec{N}$) may be represented in matrix form by its three components *a*, *b*, and *c*:(5)$$\vec{N}=\left[\begin{array}{c} a\\ b\\ c \end{array}\right]$$

The three components are also the parameters of a plane with the general form of its equation:(6)a.x+b.y+c.z=d

This plane represents the eigenplane of the crystallite. Its parameter *d* is calculated as the dot product between the normal vector matrix and a row matrix containing the averaged *x*, *y*, and *z* coordinates of the *C_8_* atoms in the crystallite, i.e., the averages of each row of **M**. Examples of two planes obtained in this way are given in Figure 4. The drawing at the top represents a plane fitted to a crystallite where the *C_8_* atoms lie almost perfectly in the same plane, whereas in Figure 4B, the crystallite features a defect. In both cases, the procedure is able to compute the correct plane.

As mentioned above, the tilt angle (θ) of the molecules in the crystallite with respect to the eigenplane is one of the order parameters for determining the phase state of a frozen system [2]. The tilt angle of each molecule in the crystallite is defined as the angle between the vector $\vec{C_{3}C_{13}}$ spanning the molecule (Figure 2A) and the normal of the eigenplane (Figure 5). The vector $\vec{C_{3}C_{13}}$ is selected because it is parallel to the molecular long axis when the alkyl chain is in *all-trans* conformation.

θ (in rad) is then calculated by the formula:(7)$$\theta =\arccos\frac{\vec{N}.\vec{C_{3}C_{13}}}{\left|N\right|.\left|C_{3}C_{13}\right|}$$

We compute the evolution of θ for each molecule in a crystallite in every frame of the trajectory, which could then be averaged over time. In addition, the values could also be averaged in each frame over all molecules. Thus, both temporal and spatial information is obtained. The standard deviations of the latter average could be used to assess the mobility of the molecules in a given crystallite. The former standard deviation could serve as a measure of the degree of ordering of the entire crystallite over time.

Similar angles were estimated by Ryckaert et al. [11] with a formula such as the one shown in Equation (1). When calculated in the way suggested in Ref. [11], however, the tilt angles are dependent on the Cartesian coordinate system and would reflect the tilt of the molecules relative to the eigenplane of the crystallite only if this eigenplane coincides with one of the planes of the coordinate system. In our case, this requirement is alleviated, which renders the procedure more universal.

A sample outcome is shown in Figure 6. Both the time-averaged tilt angle (Figure 6A) and the evolution of the molecule-averaged tilt along the last 100 ns (Figure 6B) reveal a stable average value (of all molecules in the crystallite and with time) with a standard deviation of about 2°. This result indicates low mobility of the molecules as expected in a crystal. However, the average value of 15° suggests that the molecules in the crystallite are in a state which is transient between a rotator phase with a tilt angle around zero and the more stable crystalline structure because the molecular tilt angle in the triclinic crystal of hexadecane is 19.4° [1].

### 2.4. Preparing the Crystallite for 2D Voronoi Tessellation

Other important parameters of solid-state packing are the number of nearest neighbors of a given molecule and the average area the molecule occupies. As far as rotator phases are concerned, the distances to the first neighbors are essential as well [3]. They could be used to quantify the degree of deformation of the unit cell. Performing 2D Voronoi tessellation [21,22] on the crystallite provides all the necessary information. However, this procedure requires all key points, employed to determine the polygon of each molecule, to be in the same plane. This is not strictly fulfilled for any atom or for the COM of the molecules in a crystallite, because of the thermal motion in the direction normal to the eigenplane. This type of motion is even more pronounced in rotator phases. Thus, to derive the key points (one per alkyl chain) located in one plane and at the same time pertaining to the molecular position, we calculate the point of intersection between the $\vec{C_{3}C_{13}}$ vector of each molecule in the crystallite and the crystallite eigenplane. To do that, we construct the parametric equation of the line extending along the vector $\vec{C_{3}C_{13}}$. Then, a point on the line with coordinates $\left(\begin{array}{ccc} x, & y, & z \end{array}\right)$ can be calculated as:(8)$$\begin{array}{c} x=C_{3_{x}}+l.\left(C_{13_{x}}-C_{3_{x}}\right)\\ y=C_{3_{y}}+l.\left(C_{13_{y}}-C_{3_{y}}\right)\\ z=C_{3_{z}}+l.\left(C_{13_{z}}-C_{3_{z}}\right) \end{array}$$
where *C_3x_*, *C_3y_*, *C_3z_*, *C_13x_*, *C_13y_* and *C_13z_* are the coordinates of the *C_3_* and *C_13_* atoms of the molecule and *l* is a parameter with which one can obtain every point on the line. Substituting the *x*, *y*, and *z* expressions from Equation (8) in Equation (6), we calculate the parameter *l* for the intersection between any plane and any molecular vector. In the final form, the equation is as follows:(9)$$l=\frac{d-a.C_{3_{x}}-b.C_{3_{x}}-c.C_{3_{z}}}{a.\Delta C_{3-13_{x}}+b.\Delta C_{3-13_{y}}+c.\Delta C_{3-13_{z}}}$$

Substituting *l* back in Equation (8) produces the coordinates of the point of intersection of the $\vec{C_{3}C_{13}}$ vector with the eigenplane of the crystallite. An illustration is provided in Figure 7.

For convenience, we also rotated the points to lie in the *xy* plane of the coordinate system. This was done by applying a rotation matrix on the coordinates of the intersections. To construct the matrix, we use the angle between the point of intersection (considered as a radius vector in this case) and the $\left[\begin{array}{ccc} 0, & 0, & 1 \end{array}\right]$ vector ($\vec{z}\;unit\;vector$). We then calculate their cross product, which gives the axis of rotation. The corresponding rotation matrix [23] has the form:(10)$$R=\left(\begin{array}{ccc} x^{2}.p+q & x.y.p-z.s & x.z.p+y.s\\ y.x.p+z.s & y^{2}.p+q & y.z.p-x.s\\ z.x.p-y.s & z.y.p+x.s & z^{2}.p+q \end{array}\right)$$
where *x*, *y*, and *z* are the components of the axis of rotation, q=cosϕ (*ϕ* is the angle closed between the point of intersection and $\vec{z}$), s=sinϕ, and p=1−q.

The coordinates of the intersections are obtained for each molecule and for each time frame and saved in a two-dimensional array in the format (time, *x_1_*, *y_1_*, *z_1_*, *x_2_*, *y_2_*, *z_2_*, …). The latter could then be subject to standard 2D Voronoi analysis [22,23].

Sample distances between the nearest-neighbor molecules in a crystallite of the HEX/Surf/water system simulated at 289 K, obtained from Voronoi analysis, are given in Figure 8A,B. As evident from the plots, the average distance between the nearest neighbors is much more similar to the reference plot on the right. There, the reference system was constructed [19] in a R_I_ phase. However, after 500 ns of MD simulation, it underwent a partial transition towards a more ordered crystalline phase. Comparing the reference rotator phase with the crystallite of the HEX/Surf/water system reveals the same placement of the primary and secondary peaks. This is an indication that the model system is experiencing a similar transition from the rotator to a more ordered crystalline phase. This is further confirmed by the RDF profiles presented in Figure 8C,D. The functions are calculated between the centers of mass of the molecules in the crystallite. The main difference between the reference rotator phase and the model system is the intensity of the two satellite peaks. However, the positions and the spacing of the peaks are the same. This indicates that the HEX/Surf/water is at an earlier stage of the transition than the reference model.

### 2.5. Calculating the Orientation of the Molecules in a Plane

Part of the identification of rotator or crystalline phases is the significant rotation about the long molecular axis in the rotator phases [14]. The true crystals feature uniform rotational order of all molecules. This rotation may be quantified by calculating the tensor of the moments of inertia. Then, the second principal axis (P_2_) for each molecule in the crystallite reveals the orientation of the molecules therein in a plane perpendicular to their long axis (Figure 9A). This approach was recently applied successfully for the characterization of rotator phases in bulk alkanes [18].

Evaluating the principal axes as eigenvectors of the tensor of the moments of inertia for a single molecule produces an orthogonal basis where the vector with the highest eigenvalue is parallel to the longest axis of the molecule and the vector with the second-highest eigenvalue describes the direction in which the bonds are oriented (Figure 9B). Thus, if the molecules are rotated so that their long axes are parallel to *z*, the second principal axis reveals the orientation of the molecule in the *xy* plane.

In our approach, we first rotate the crystallite by an angle ensuring that the molecules are parallel to the *z* axis of the coordinate system, as explained in Section 4. After that, we calculate the second principal axis of each molecule in the crystallite and its angle with respect to the *x* and *y* axes of the coordinate system. This is done for every *n*-th frame in the trajectory producing a movie of consecutive plots (vector fields as those shown in Figure 9), which reveals the orientation of the molecules as a function of time (Movies S1–S3 of the Appendix A). The vectors are always centered at the COM coordinates of the respective molecule from the last frame of the trajectory. A movie of this series of plots (e.g., Movie S1) visualizes the evolution of the intermolecular orientation. It should be noted that the movie may be started from the liquid state and encompass the whole trajectory. Hence, the formation of all phases during and after freezing is illustrated directly. This is shown on the examples of the single crystal of HEX/Surf/water at 289 K (Figure 9B,C, Movie S1), crystallite 3 (Figure 9D,E, Movie S2), and crystallite 7 (Figure 9F, Movie S3) of the same system at 278 K. The crystallites are labeled in Figure 1C.

The P_2_ analysis of HEX/Surf/water at 289 K reveals that a rotator phase is formed right after freezing (Movie S1), which then spontaneously and collectively converts to a more ordered (crystalline-like) structure. The two ordered states remain in equilibrium until the end of the simulation, but the crystalline-like state is more stable, judging from its more prevalent population.

Crystallite 3 of HEX/Surf/water at 278 K is in a quasi-stable state with both phases described above equally represented. Only in the last 30 ns a noticeable shift toward the crystalline-like phase is observed (Figure 9E).

Crystallite 7 of HEX/Surf/water at 278 K, which is one of the surfactant-rich surfaces, features a different type of packing compared to the rest of the crystallites, which is stable throughout the entire trajectory (Figure 9F). The prevalence of surfactants on the surface warrants further investigation to evaluate the phase state of this structure.

Overall, the movies with the evolution of P_2_ orientation are an indispensable part of the analysis toolbox needed for the assignment of the type of solid-state packing.

### 2.6. Overview of the Analyses of the HEX/Surf/Water System at 289 K

An illustrative application of the analysis procedure described in this paper was done for the model system HEX/Surf/water (Figure 1D) simulated at 289 K. In summary, the calculated tilt angles, nearest-neighbor distances, radial distribution functions, and P_2_ orientation are in agreement that the system is experiencing a transition from a rotator phase to a more ordered crystalline state. The average tilt angle of the system is ~15° (Figure 6), which is greater than the one expected for a rotator phase [3] and lower than that determined for a triclinic crystal [1]. Comparing the distances of the nearest neighbors in the crystallite and the RDFs with a reference rotator phase (the rotator phase underwent a transition to a more ordered crystalline state) and with a reference triclinic phase revealed much more similarities with the former. The P_2_ analysis provided a useful visual confirmation of the initial transition of the system from liquid to rotator phase and then to a more ordered crystalline state, still co-existing with a rotator-type arrangement.

The combination of those analyses provided a definitive evaluation of the structuring in the obtained crystallite, as explained above in Section 2.1–Section 2.5.

## 3. Materials and Methods

### 3.1. Model Systems

The methodology described in the current paper is developed and tested on two types of hexadecane-containing systems (Figure 1). The first one (Figure 1A,B) is bulk hexadecane (bulk HEX) represented by a box of 440 molecules and the second model (Figure 1C,D) consists of 494 molecules HEX, 108 surfactant molecules (2-(2-(hexadecyloxy)ethoxy)ethanol, C_16_(EO)_2_) and 2778 water molecules (HEX/Surf/water) arranged at a flat surfactant-stabilized oil-water interface. Both models are placed in a hexagonal periodic box with initial dimensions of 5.2 × 4.5 × 9 nm for bulk HEX and 5.2 × 4.5 × 17.2 nm for the HEX/Surf/water system. In the starting configuration, the molecules are placed on the nodes of a regular lattice with an initial area per molecule of 0.195 nm^2^ [3,24] and each molecule is randomly rotated about its long axis.

### 3.2. Simulation Protocol

The two model systems are simulated with atomistic molecular dynamics using the force field CHARMM36 [25] for the hexadecane and surfactant, and the potential TIP4P [26] to describe water. First, the model systems are melted to isotropic liquid by heating them to 350 K and then cooled down to 300 K. The liquid is equilibrated at this temperature for 200 ns. After that, the systems are cooled down to a temperature (278 K) where a phase transition to an ordered phase is observed. Then, a 500 ns simulation is conducted in the NPT ensemble at 278 K [27] and 1 bar [28]. The pressure scaling is isotropic in bulk HEX and semi-isotropic [29] in HEX/Surf/water. In the latter, the scaling of the *z*-axis of the box is decoupled from that of *x* and *y*. The frozen models are polycrystalline in nature (Figure 1). It should be noted that 278 K was the highest temperature at which the model systems froze after cooling. This is ca. 13 K lower than the experimentally measured freezing point of hexadecane [3,30]. To check whether this deviation is an artifact of the used force field or just supercooling due to the nanosized models [3], an additional simulation for HEX/Surf/water is performed (Figure 1D). This is done to evaluate the effect of the nanoconfinement of the system on the freezing process. In this run, the starting structure is taken from the trajectory simulated at 278 K after a nucleation seed was formed. The molecules are then heated up to 289 K, which is close to the experimental freezing temperature of the system. Apart from the different final temperature, the same simulation protocol after cooling is followed as for the other two simulations.

The MD computations and some of the analyses are done with GROMACS 2020 [31]. Post-processing of the trajectories is carried out with the framework MDAnalysis 2.1.0 [32,33] and with NumPy 1.22.3 [34]. VMD 1.9.4 [35] and Matplotlib 3.5.2 [36] are used for visualization.

## 4. Conclusions

In this work, we present a comprehensive methodology for the analysis of hexadecane-containing systems that have undergone a phase transition from isotropic liquid to an ordered solid phase within a molecular dynamics simulation. The described analyses are developed with the intent to quantify rotator and crystalline phases in alkane-based systems. However, the procedure is more versatile and can be utilized for the characterization of other solid-state structures comprised of quasilinear molecules, including monolayers and bilayers of lipid molecules.

Most solid-state systems feature a polycrystalline structure. Thus, a reliable method to isolate the individual crystallites is proposed. An additional routine for post-processing of the crystallites is developed, which enables obtaining their real-space coordinates from periodic-box-imaged constructions. This is followed by a method to fit the eigenplane of each crystallite and to determine the coordinates of the points of intersection of each molecule in the crystallite with the eigenplane. These basic structural units are then used to calculate the molecular tilt angles in the crystallites and to define the type of intermolecular packing by Voronoi analysis.

The orientation of the second principal axis calculated from the tensor of the moments of inertia is estimated. Its temporal and spatial evolution is proposed as a sensitive indicator to reveal the transitions between various phases present in the systems, e.g., from a liquid to a rotator phase and then to a crystalline one.

As pointed out throughout the paper, several of the proposed analyses offer an improvement over previous studies [9,10,11,12,13,14,15,16,17,18]. The main advantage of our procedure is that the analyses are rendered independent of the Cartesian coordinate system. This feature allows one to process solid-state assemblies where multiple crystallites (domains) are formed in the same model.

The modular construction of the analysis procedure allows its straightforward extension to other structural characteristics or other chemical entities. This is facilitated also by employing only open source and generally popular programming tools. The scripts are available under the general public license (v. 2.0) at: https://github.com/Iliev-S/crystallite_analysis (accessed on 28 February 2023).

## Figures and Tables

**Figure 1 molecules-28-02327-f001:**
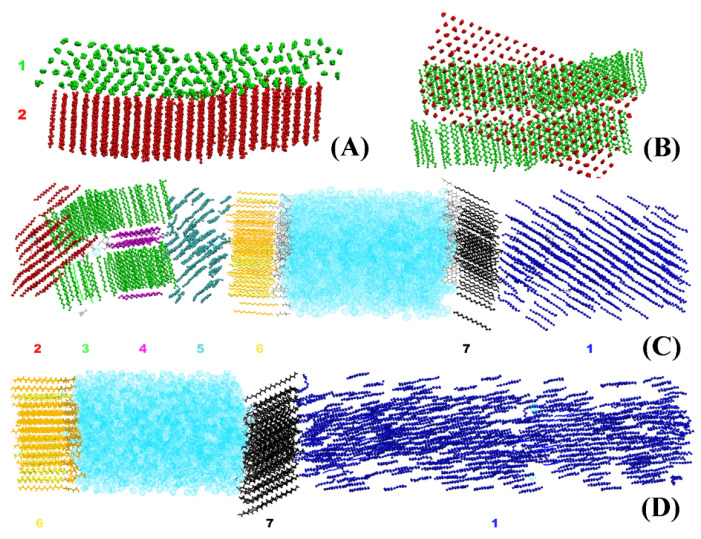
Illustration of the crystallites in the model systems after 500 ns simulation in the solid state: (**A**) side and (**B**) top view of the crystallites in bulk HEX, side view of all crystallites in HEX/Surf/water (**C**) at 278 K and (**D**) at 289 K. Each color represents a separate crystallite, and each crystallite is assigned a number. Hexadecane is represented with beads, the surfactant tails—with thick lines, the surfactant heads—with thin lines, and the water molecules—with bubbles.

**Figure 2 molecules-28-02327-f002:**
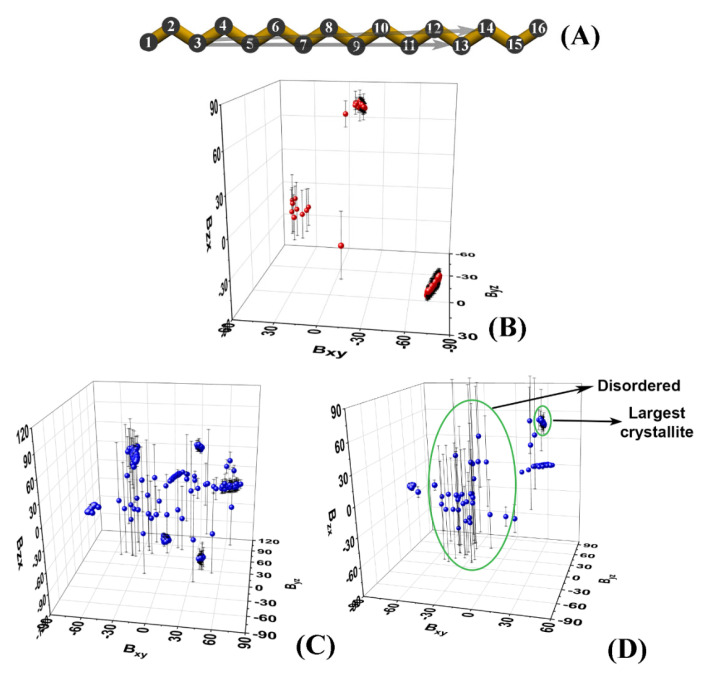
(**A**) Carbon skeleton of hexadecane with numbering of the atoms; the two arrows denote the vectors $\vec{C_{3}C_{13}}$ and $\vec{C_{3}C_{14}}$ used for the analyses in this paper; 3D plots of the slopes of the vectors $\vec{C_{3}C_{14}}$ averaged over the last 50 ns of the MD trajectories in (**B**) bulk HEX simulated at 278 K, (**C**) HEX/Surf/water simulated at 278 K, and (**D**) HEX/Surf/water simulated at 289 K.

**Figure 3 molecules-28-02327-f003:**
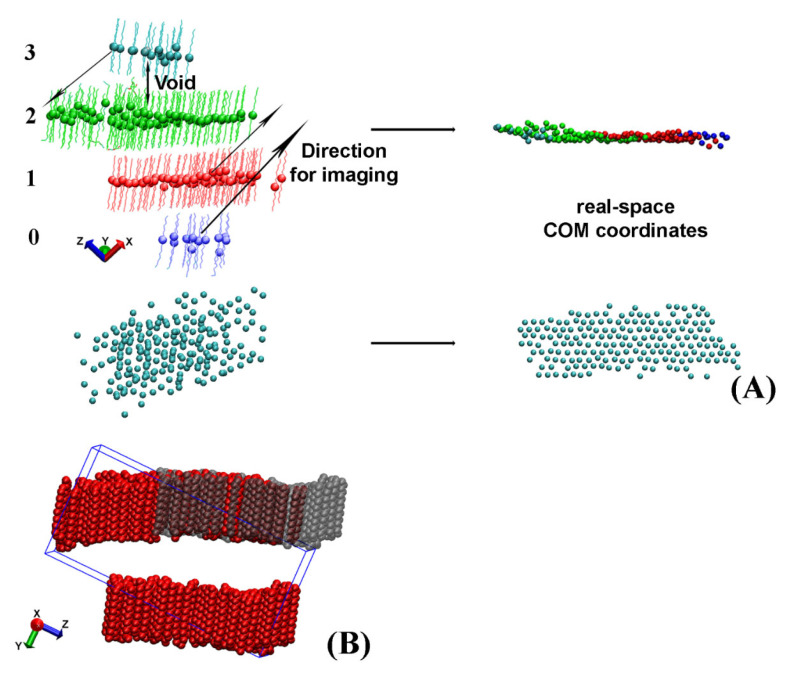
(**A**) Scheme of the procedure employed to reconstruct the real-space coordinates of a crystallite split across the periodic box taken from the HEX/Surf/water system. Each bead represents the center of mass (COM) of a molecule in the crystallite. The numbers on the left of the figure are indices of the slabs. In the given example, the direction of imaging is along *x*. (**B**) Illustration of the largest crystallite in bulk HEX split across the periodic box in the *y* direction. The red molecules are those saved in the periodic box and the grey ones are the ones resulting from the imaging along *y* to their real-space coordinates.

**Figure 4 molecules-28-02327-f004:**
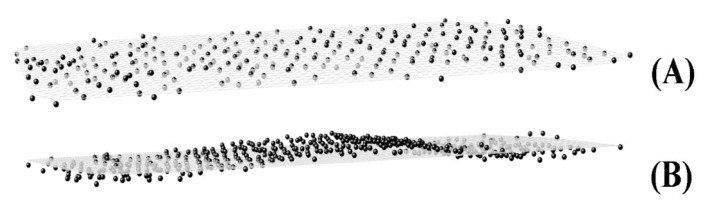
Fitted plane to the middle (*C_8_*) atoms in a crystallite of HEX/Surf/water simulated at 289 K (**A**) without a defect and (**B**) with a defect.

**Figure 5 molecules-28-02327-f005:**
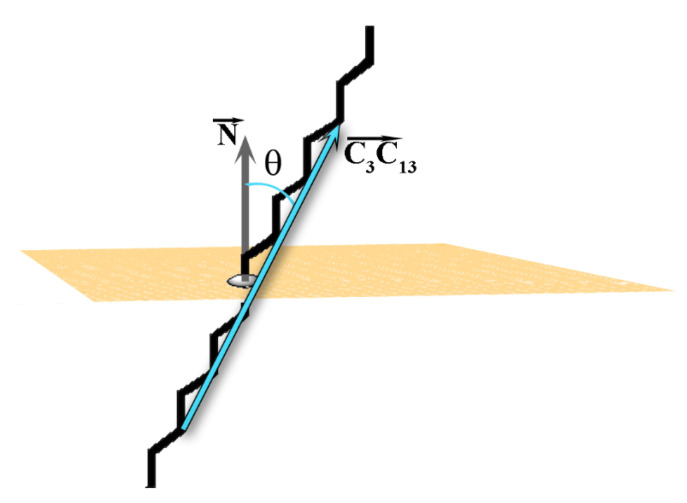
Illustration of the tilt angle θ. The grey arrow is the normal vector of the plane ($\vec{N}$
) and the cyan arrow is the vector $\vec{C_{3}C_{13}}$ constructed from the third and thirteenth carbon atoms of each molecule in the crystallite.

**Figure 6 molecules-28-02327-f006:**
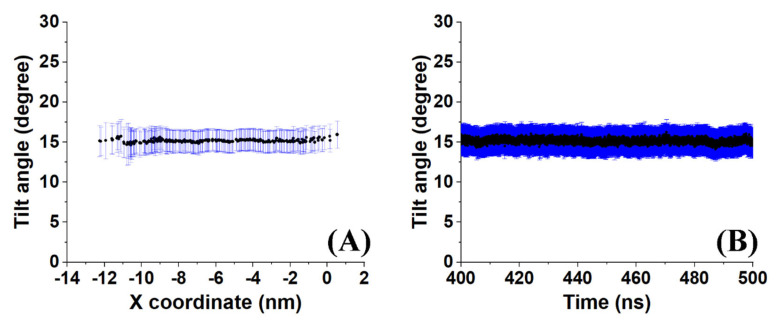
Illustration of (**A**) the time–averaged tilt angle of the molecules in a crystallite of HEX/Surf/water simulated at 289 K and (**B**) the evolution of the average tilt angle along the last 100 ns of the trajectory.

**Figure 7 molecules-28-02327-f007:**
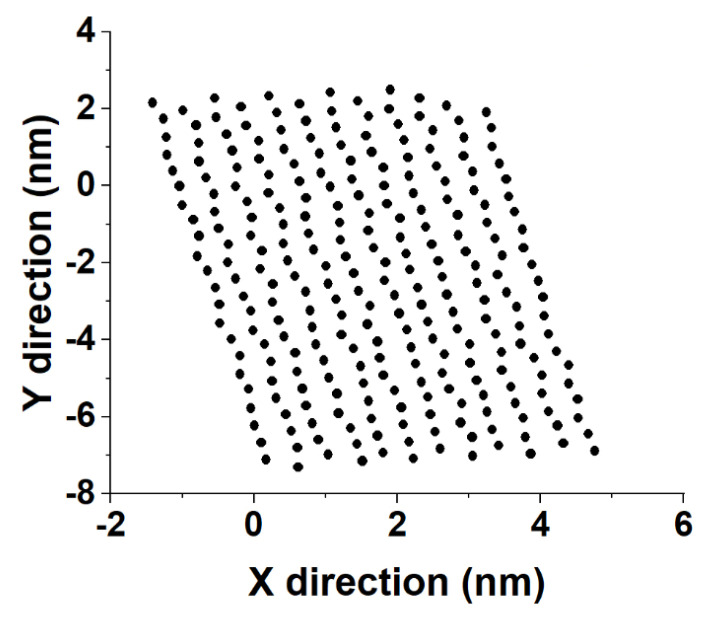
Illustration of the obtained in–plane coordinates of the intersection points of alkyl chains with the plane of crystallite 2 in the bulk HEX system at 278 K in the last frame of the MD trajectory.

**Figure 8 molecules-28-02327-f008:**
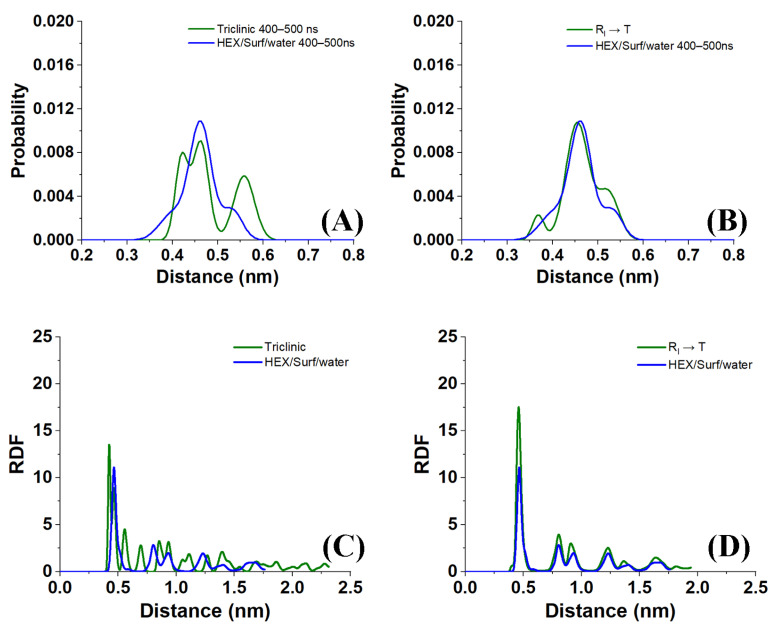
(**A**,**B**) Distances between the nearest neighbors and (**C**,**D**) RDF profiles in crystallite 1 of the HEX/Surf/water system simulated at 289 K compared with reference systems [19] of bulk HEX in (**A**,**C**) triclinic phase and (**B**,**D**) rotator phase. All data in the graphs were obtained by averaging over the last 100 ns of a 500 ns MD production run.

**Figure 9 molecules-28-02327-f009:**
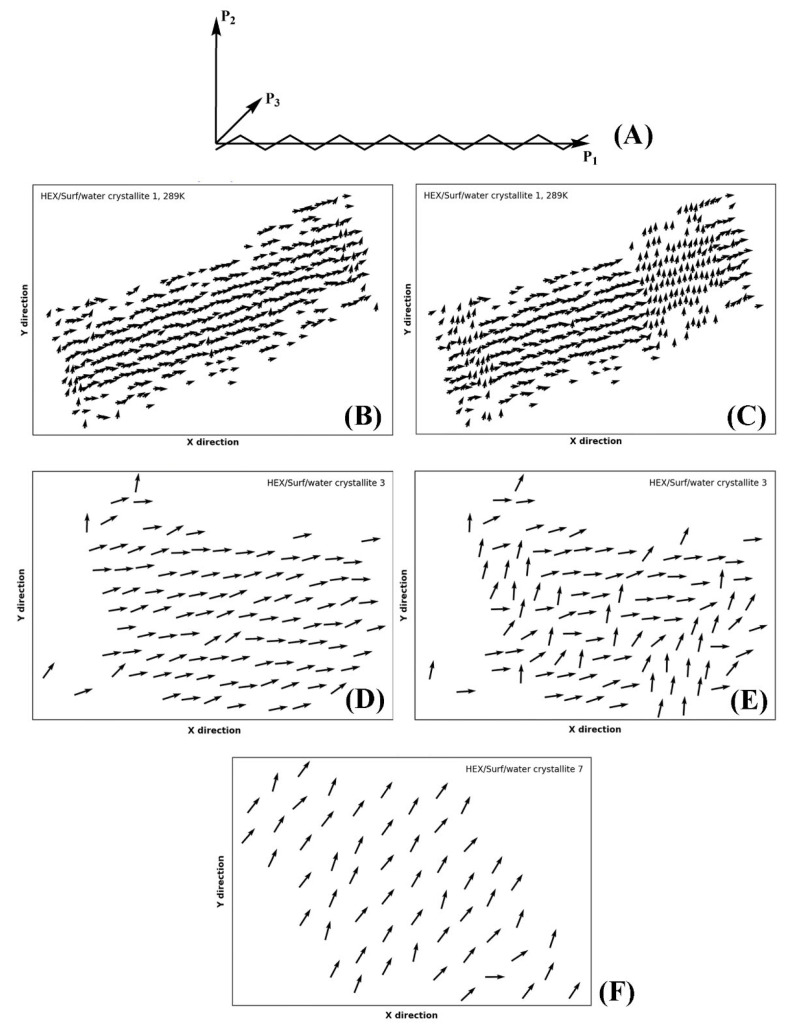
(**A**) Illustration of the three principal axes describing the orientation of the molecule in space; P_1_ is aligned with the long molecular axis, P_2_ is orthogonal to P_1_ and gives insight into the rotation of the molecule about P_1_ in space, and P_3_ is orthogonal to both P_1_ and P_2_; orientation of the axis P_2_ of each molecule in crystallite 1 of HEX/Surf/water simulated at 289 K, in (**B**) crystalline-like state, and (**C**) both ordered states present in the crystallite at once; crystallite 3 of HEX/Surf/water at 278 K in (**D**) the crystalline-like state, and in (**E**) both ordered states in equilibrium; (**F**) crystallite 7 of HEX/Surf/water at 278 K.

## Data Availability

Not applicable.

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
