# Peer review of "Computational Procedure for Analysis of Crystallites in Polycrystalline Solids of Quasilinear Molecules"

_molecules, 2023, doi:10.3390/molecules28052327_

Round 1
Reviewer 1 Report
In this paper, Ivanova et al. present a comprehensive procedure for structural analysis of quasilinear organic molecules arranged in a polycrystalline sample generated by molecular dynamics. The work is solid and should be of interest to researchers in the filed of solid state structures. The manuscript has been well organized. I would suggest acceptance of this paper by Molecules upon minor revisions.
1. The use of citation format is somewhat odd. Please adjust it to a better manner.
2. Some sentences can be optimized. For example,
1) Modelling of the freezing or melting of alkane-based systems with molecular dynamics (MD) can provide insight into the nature of formation of the intermediate phases and describe transformations between transient phases too short 44 lived to be observed experimentally.
2) Another calculated parameter ws the orientation of the molecules 65 with respect to the x axis of the coordinate system.
3) There, also in-depth structural analyses of these systems are made by using the procedures developed in the current study.
4) “In contrast” or “By contrast”?
3. “eq. 1a” et al. should be well aligned with corresponding equations.
Author Response
A file with the reply is attached.

Reviewer 2 Report
The paper of Iliev and coauthors provides a detailed account of a new approach to the analysis of the transient solid-state structures (known as rotator phases) made of organic chains. The approach is based on molecular dynamics methods and is deemed as a universal protocol of studying polycrystalline systems made of quasilinear organic molecules. The paper is well written, it fits to the scope of the journal nicely, and can be published subject to minor revision (please see minor comments below):
1. Lines 246-252; the authors remark that the scalar product gives the eigenplane parameter d and then give two examples of such planes, with and without defects. However, it is unclear how the scalar product accounts for defects. I believe that the authors ought to give more details/explanations.
2. The reference numbering is a bit unusual. Why not using typical Arabic numerals for references?
Author Response
A file with the reply is attached.
